# The Graphical Representation of Cell Count Representation: A New Procedure for the Diagnosis of Periprosthetic Joint Infections

**DOI:** 10.3390/antibiotics10040346

**Published:** 2021-03-24

**Authors:** Bernd Fink, Marius Hoyka, Elke Weissbarth, Philipp Schuster, Irina Berger

**Affiliations:** 1Department for Joint Replacement, Rheumatoid and General Orthopaedics, Orthopaedic Clinic Markgröningen, Kurt-Lindemann-Weg 10, 71706 Markgröningen, Germany; m.hoyka@web.de (M.H.); elke.weissbarth@rkh-kliniken.de (E.W.); philipp.schuster@rkh-kliniken.de (P.S.); 2Orthopaedic Department, University Hospital Hamburg-Eppendorf, Martinistrasse 52, 20251 Hamburg, Germany; 3Department of Orthopedics and Traumatology, Clinic Nuremberg, Paracelsus Medical Private University, Nuremberg, Breslauer Straße 201, 90471 Nürnberg, Germany; 4Department of Pathology, Klinikum Kassel, Mönchebergstraße 41-43, 34125 Kassel, Germany; irina.berger@klinikum-kassel.de

**Keywords:** periprosthetic joint infection, diagnosis, leukocyte, cell count

## Abstract

Aim: This study was designed to answer the question whether a graphical representation increase the diagnostic value of automated leucocyte counting of the synovial fluid in the diagnosis of periprosthetic joint infections (PJI). Material and methods: Synovial aspirates from 322 patients (162 women, 160 men) with revisions of 192 total knee and 130 hip arthroplasties were analysed with microbiological cultivation, determination of cell counts and assay of the biomarker alpha-defensin (170 cases). In addition, microbiological and histological analysis of the periprosthetic tissue obtained during the revision surgery was carried out using the ICM classification and the histological classification of Morawietz and Krenn. The synovial aspirates were additionally analysed to produce dot plot representations (LMNE matrices) of the cells and particles in the aspirates using the hematology analyser ABX Pentra XL 80. Results: 112 patients (34.8%) had an infection according to the ICM criteria. When analysing the graphical LMNE matrices from synovia cell counting, four types could be differentiated: the type “wear particles” (I) in 28.3%, the type “infection” (II) in 24.8%, the “combined” type (III) in 15.5% and “indeterminate” type (IV) in 31.4%. There was a significant correlation between the graphical LMNE-types and the histological types of Morawietz and Krenn (*p* < 0.001 and Cramer test V value of 0.529). The addition of the LMNE-Matrix assessment increased the diagnostic value of the cell count and the cut-off value of the WBC count could be set lower by adding the LMNE-Matrix to the diagnostic procedure. Conclusion: The graphical representation of the cell count analysis of synovial aspirates is a new and helpful method for differentiating between real periprosthetic infections with an increased leukocyte count and false positive data resulting from wear particles. This new approach helps to increase the diagnostic value of cell count analysis in the diagnosis of PJI.

## 1. Introduction

Periprosthetic joint infection (PJI) is a devastating complication of arthroplasty procedures and has many consequences. The level of incidence for total hip and knee arthroplasties ranges between 1% and 2% on average [1]. However, in some reports this type of infection is claimed to be the most frequent cause of implant failure during the first five years following surgery [2,3,4]. Thus, the accuracy of the preoperative diagnosis of possible periprosthetic joint infection becomes especially important in cases of loosened or painful endoprostheses [5,6].

Whereas early infections, i.e., those occurring within the first four weeks of implantation, usually cause local and systemic inflammatory reactions, these signs are often missing in cases of late PJI. This makes the diagnosis of late periprosthetic infections very much more difficult. 

An important diagnostic method for late PJI is the determination of the leukocyte count (WBC) in the joint synovia. Some authors consider it one of the most important diagnostic parameters [7,8] and it is one of the criteria in the definition of the PJI, the MSIS criteria and the more recent ICM criteria [9,10,11]. However, the cut-off value given in the literature for the leukocyte number that correlates with a positive PJI differs considerably between investigators, with a range from 1100 to 5000 cells/µL (Table 1). A reason for this could be that factors such as the time elapsed since the operation, the duration of the symptoms, the causative microorganism, previous antibiotic use and co-morbid conditions all seem to influence the results [12,13,14,15]. On the other hand, when determining the cell count, wear particles from the articulation surface of the joint (polyethylene particles or metal particles) may also be counted, which incorrectly increases the final cell count measured in the cell counter [16,17,18]. This could explain why Deirmengian et al. [18] found an increased risk of false-positive automated synovial fluid WBC counts from hip and knee arthroplasties. This phenomenon applies above all to metallic wear particles that arise from metal-on-metal articulations and metal-on-polyethylene articulations with corrosion. Here metal particles in the joint aspirate have been reported to produce falsely high leukocyte values during cell counting, an increased serum CRP-level and a positive alpha-defensin assay in at least one third of cases [17,19,20,21,22,23,24]. In addition, aspirates that resemble pus can occur in metal-metal pairings, which make it difficult to differentiate between aspirates associated with joint infection and those containing metal wear particles [20,21,22,23,24]. Therefore, especially when wear particle debris is accompanied by an increased serum CRP-level and positive alpha-defensin in the aspirate, or even one positive culture, exact counting of the leukocytes in the aspirate is necessary to distinguish between wear debris (leading to less than six points in the ICM-criteria [10]) and a real periprosthetic joint infection (with at least six points in the ICM-criteria [10]). Therefore, an automated synovial WBC counting that can differentiate between wear particles and raised numbers of leukocytes due to periprosthetic joint infection would be helpful to diagnose and treat these patients correctly. 

The different volume and the different behaviour with respect to light absorption means that wear particles and leukocytes can be differentiated using a graphic representation of automated cell counting of synovial fluid. The data can be represented in a graphical dot-plot display (LMNE-matrix). The wear particles will be found in the so-called NOISE area (area of impurities) of these graphical representations (Figure 1). The analysis of the aspirate would therefore produce different images according to the content of the synovial fluid: the pure wear particle type where particles were present in significant numbers, the pure infection type with high neutrophil counts, the combined type with high counts of both neutrophils and wear particles, and possibly an indeterminate type with no clear distribution of either cells or particles. A similar classification was developed by Morawietz and Krenn [36,37,38] for the histological evaluation of the periprosthetic tissue. Therefore, this histological evaluation and histopathological consensus classification of periprosthetic membranes could function as a control for the differentiation of the different graphical types obtained from the cell count analysis [37].

Thus, the objectives of the present study were to answer the following questions: When determining the cell count of the aspirate, can the different types of dot-plot image be distinguished?

Do those types correlate with the types of histology characterized by Morawietz and Krenn [36,37,38]?

Does the graphic representation of the cell count help to increase the diagnostic value of the cell count measurement of the synovial fluid?

## 2. Results

112 patients (34.8%) had an infection according to the ICM criteria. When analysing the LMNE matrices of the cells in the synovial fluid, four types could be differentiated. First, there were clusters of data points at the border to, and in the NOISE area of, the LMNE matrix, which could not be assigned to any cell type and were identified as wear particles in subsequent tests (Figure 2). Analysing polyethylene wear particles produced in the laboratory in Ringer’s solutions, revealed that they were associated with the NOISE area at the top of the LMNE matrix, on the left (Figure 3a). In the case of clinically unambiguous and macroscopically visible metal wear particles (with articulating ceramic heads in hip prosthesis cups due to defective polyethylene inlays), the metal abrasion products were mostly located in the lower left of the LMNE matrix at the border to the NOISE-area and showed an “L”-shaped distribution (Figure 3b). Based on the classification by Morawietz and Krenn [36,37,38] for the histological classification of periprosthetic synovial tissue, this was classified as type I. 91 patients (28.3%) exhibited this type I LMNE matrix. A cluster of data points that corresponds to the position of neutrophil leukocytes in the graphical representation corresponded to infection type II (80 patients, 24.8%) (Figure 4). If the wear particle levels as well as the neutrophil leukocyte levels were high, this was designated as the combined type III (50 patients, 15.5%) (Figure 5). All other LMNE matrices that did not show a clear differentiation of cell types or particles were classified as the indeterminate type IV (101 patients, 31.4%) (Figure 6).

Thus 130 aspirates (40.4%) were associated with an infection (type II and III) in the LMNE matrix analysis (Table 2). Comparing the evaluation of the LMNE-matrices with the histological types according to Morawietz and Krenn [36,37,38], there was a significant correlation of *p* < 0.001 for the chi-square test and a Cramer test V value of 0.529. Table 3 shows the diagnostic value of the various tests taking into account the ICM criteria. It was found that the addition of the LMNE-matrix evaluation increased the diagnostic value of the cell count and the threshold value of the WBC count could be set lower by considering the diagnostic significance of the LMNE-matrix as well (Table 3). The calculation of the cell count threshold using the receiver operating characteristic curve analysis resulted in a cut-off of 1400 cells/µL at a sensitivity of 90.2% and a specificity of 91.9% (Figure 7).

## 3. Discussion

There was a significant level of agreement between the four distribution types in the LMNE matrix in the cell count analysis and the four histopathological types described by Morawietz and Krenn [36,37,38]. Thus, the type classification we have chosen seems to agree with the other diagnostic methods. This in turn helps to distinguish between a real infection and a wear debris type where the cluster shown on the cell counter matrix is due to wear particles and not to actual leucocytes. Above all, this method can be used to differentiate leukocytes from metallic abrasion particles, which appears to be of particular importance, since joint aspirates containing metal abrasion particles can look like pus and be associated with apparently very high cell counts, as well as exhibiting elevated CRP- and alpha-defensin values. In the absence of a graphical representation of the cell count data these features could be incorrectly interpreted as a periprosthetic infection [17,20,21,22,23,24]. 

Furthermore, the combination of cell count and LMNE types II or III enabled a lowering of the cut-off value of the cell count in the aspirate without losing its high sensitivity. Hereby, the combination of cell counting and the graphical representation in the LMNE matrix means that fewer periprosthetic infections will be overlooked and the diagnostic value of the cell count analysis in the joint aspirate is increased.

Even though this is the first description of such a type differentiation in cell count analysis, this study has some weaknesses. The number of patients was high enough to allow the significant correlations to be statistically recognized as such. Nevertheless, this description represents a first pilot study and the type differentiation proposed here must be verified by further studies with higher patient numbers. Furthermore, this type classification, as well as that in histopathology, is somewhat dependent on the personal interpretation and experience of the examiner. Even though the reliability of this type classification was very high in our study, this does not rule out a certain subjectivity in the interpretation of the LMNE matrices. In addition, the usefulness of the aspirate cell count is lessened when blood is present [39]. Even though bloody aspirates were excluded in this study according to the recommendation of Deirmengian et al. [39], even a little blood in the aspirate reduces the value of cell count measurement. In these cases, however, it is helpful that more basophils, lymphocytes and eosinophils can be found in the blood than in a joint aspirate, so that the interpretation of the data from an aspirate that contains blood is easier to recognize. Thus, when blood is contaminating the aspirate, a high number of neutrophils does not necessarily mean that there is an infection. Centrifuging the aspirate beforehand should help to make the interpretation more straightforward; this procedure significantly improved the readability of leukocyte esterase strips, for example [40]. It should be noted that such an LMNE matrix of the synovial aspirate cannot be created by all cell counting devices. This is because the creation of the LMNE matrix requires the measurement of the light absorption of the cells or particles and many cell counting devices only measure the scattered light and the size of the cells or particles. Moreover, the threshold of cell count in the ROC curve analysis at 1400 cells/µL was relatively low in our patient group. This threshold is slightly lower than that of Ghanem et al. [27], Trampuz et al. [8], DeVecchi et al. [31] and Dineen et al. [32] (Table 1) and is presumably due to the composition of the patient group that exhibited a high proportion of low-grade infections. However, this should not have any influence on the additional benefit of the LMNE matrix as a diagnostic tool.

Another possibility to distinguish leukocytes and wear particles in the aspirate involves manual counting by a laboratory technician using a microscope. However, manual counting of WBC in synovial fluid is less accurate than automated counting because it has been shown to result in an inter-observer variance of more than 20% [41,42,43,44]. Therefore, the improved automated counting procedure described here seems to be more promising than the traditional manual alternative.

Despite those weaknesses mentioned, in our opinion the graphic representation of the cell count analysis of synovial aspirates from joints with endoprostheses is a new and helpful method for diagnosing true periprosthetic infections. Using a device that graphically displays an increased leukocyte count and identifies wear particles that would otherwise lead to an incorrect interpretation of the data, will increase the diagnostic value of the cell count analysis. In our opinion, this technology should therefore be included in the diagnostic armamentarium of the orthopaedic specialist faced with cases of loosened or painful endoprostheses.

## 4. Materials and Methods

This prospective analysis included 390 patients (202 women, 188 men) who had revision surgery (212 total knee replacements, 178 total hip replacements). They all underwent a prior aspiration of the joint. Systemic inflammatory diseases such as rheumatoid arthritis were excluded because these diseases can be associated with the presence of leukocytes in the joint in the absence of a PJI [45]. Patients with a punctio sicca (dry taps) (35 hips) and 31 bloody aspirates were also excluded according to the recommendation of Deirmengian et al. [39], since the latter significantly reduce the sensitivity of cell count measurement [39]. This left 322 patients (162 women, 160 men) with revisions of 192 total knee replacements and 130 hip replacements. The mean age of the patients was 69.5 ± 10.9 years (28–95 years). The revision operation was carried out 83.1 ± 78.4 months (2–339 months) after the primary implantation. None of the patients took any antibiotics in the four weeks preceding the aspiration. The joint aspiration techniques were carried out under sterile conditions. 

Cell numbers were determined for each aspirate by pipetting at least 1 mL synovial fluid into an EDTA tube before determining the cell count with the laboratory diagnostic device, ABX Pentra XL 80 (Horiba Medical, Montpellier, France). The ABX Pentra XL 80 is a device for the analysis of the cell count and the WBC-differentiation of blood and body fluids. Here we selected the so-called 5-DIFF mode from the various processing modes available. A total of 26 laboratory parameters are recorded, including the five cell types eosinophils, neutrophils, monocytes, lymphocytes and basophils as well as atypical lymphocytes and large, immature cells. These cell types are graphically mapped in a so-called LMNE matrix, plotting their cell volume (x-axis) against their light scattering or refraction and absorption (y-axis) (Figure 1). The analysis is based on a combination of impedance measurement, flow cytometry and cytochemistry. This enables the graphical assignment and thus differentiation of the four leukocyte populations: lymphocytes, monocytes, neutrophils and eosinophils (Figure 1). Impurities—in our case, wear particles–are found in the so-called NOISE area of the LMNE matrix (Figure 1).

The evaluations and assignment of the individual matrices to the four different types of image were carried out twice by two examiners (BF and MH) independently of one another and without knowledge of the histology. It showed a high reliability, with an intrarater intraclass correlation coefficient of 0.99 and of 0.98 between raters, respectively.

Additionally, the harvested fluid was immediately aspirated into paediatric blood culture bottles containing BD BACTEC-PEDS-PLUS/F-Medium (Becton Dickinson, Heidelberg, Germany) and were incubated for 14 days [46]. In cases where enough synovial fluid was aspirated, alpha-defensin was also analysed using an ELISA-Test (170 cases). Serum CRP-levels were determined in all cases.

During the revision surgery itself, samples were taken from five different areas close to the prosthesis (synovium and periprosthetic tissue). In addition, five samples from the synovium and the periprosthetic connective tissue membrane associated with the loosened prosthesis were obtained for histological assessment. Perioperative antibiotics were only administered once all the samples had been taken. The biopsy samples were each placed in sterile tubes and transferred together with the aspirated fluid to the microbiological laboratory within an hour of sampling. The samples were streaked onto blood agar and inoculated into special nutrient broth for anaerobic organisms. All the samples were incubated for 14 days [46]. The results together with results of the aspiration were analysed according to the ICM-criteria [9,10,11]. Hereby the results were rated as periprosthetic joint infection (PJI) when the sum of the diagnostic results was at least 6. The classification by Morawietz and Krenn et al. [36,37,38] was used for the histological analysis of the periprosthetic tissue in order to differentiate between the wear particle type (I), the infection type (II), the combined type (III) and the indeterminate type (IV). In addition, the number of polymorphonuclear leukocytes per high power microscope field was also determined.

Statistical evaluation was performed using SPSS for Windows (version 22; IBM Corp.; Armonk, NY, USA). The chi-square test was used for comparison of nominal variables between groups, and Cramer-V was used for correlations between nominal variables (>0.5 was defined as strong correlation). The level of significance was generally set at *p* < 0.05. A receiver operating characteristic (ROC) curve analysis was used for calculating the cell count threshold. Sensitivity and specificity as well as likelihood ratios were calculated in order to evaluate the performance of tests and to choose a diagnostic threshold that is based on the best combination of sensitivity and specificity.

## Figures and Tables

**Figure 1 antibiotics-10-00346-f001:**
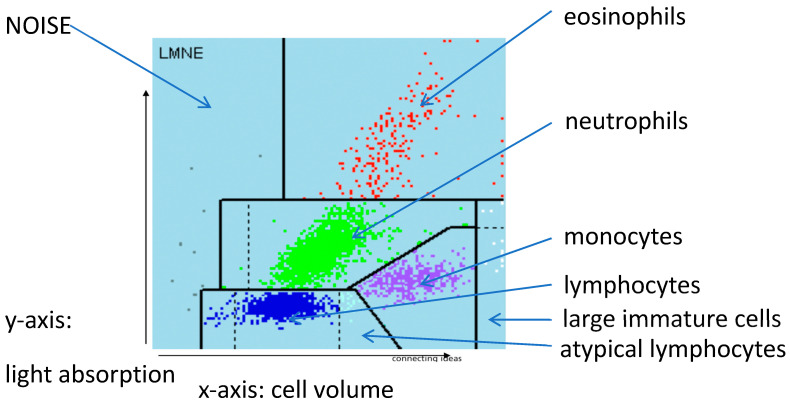
LMNE matrix with the different fields for the leukocyte populations and the NOISE area.

**Figure 2 antibiotics-10-00346-f002:**
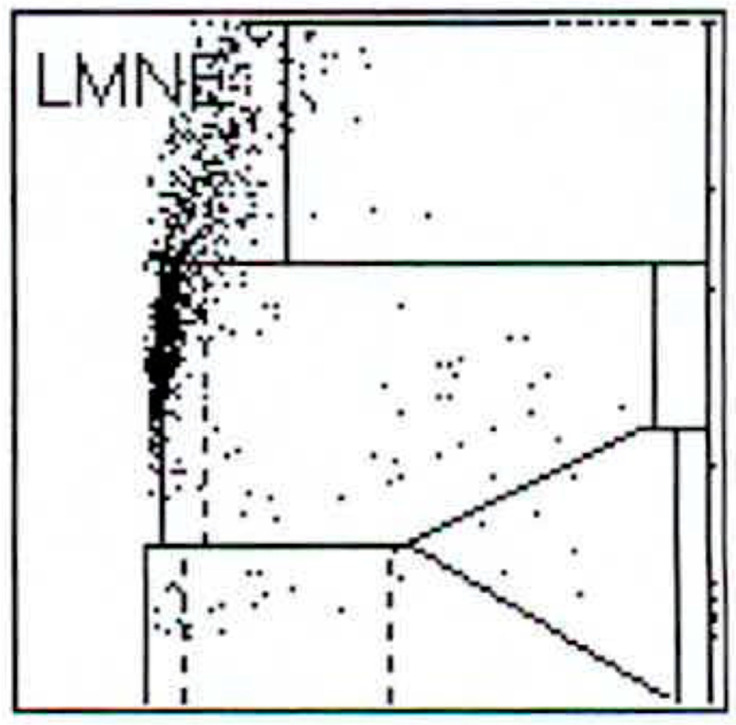
LMNE matrix of a type I (abrasion type) with a cloud in the NOISE-area of a 65-year-old male patient with an aspirate of the hip arthroplasty 15 years postoperative. The measured cell count was 1500 cells/µL.

**Figure 3 antibiotics-10-00346-f003:**
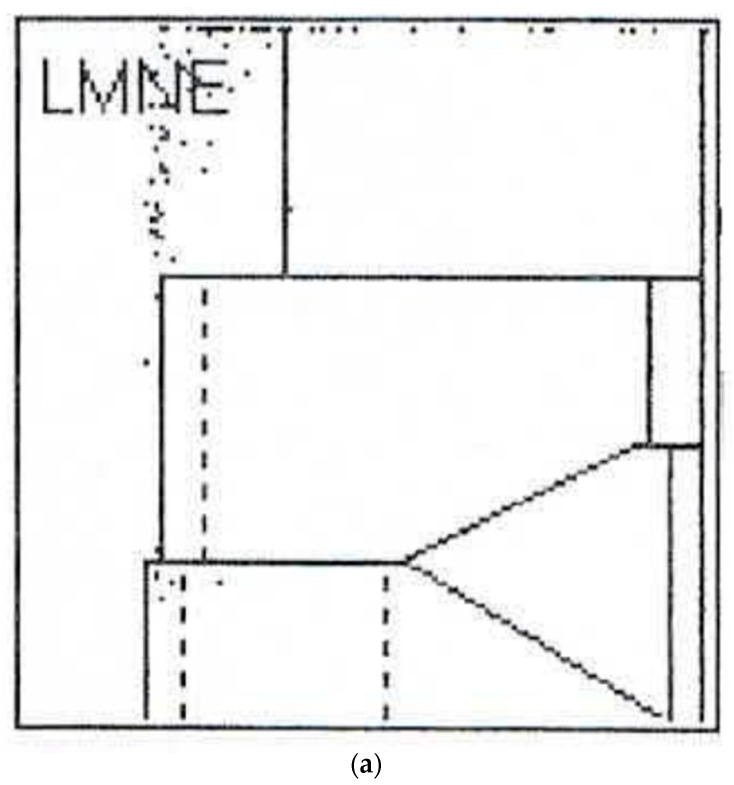
(**a**) LMNE matrix of a type I with polyethylene wear particles produced in a laboratory. The cloud is at the top in the NOISE area. (**b**) LMNE matrix of a type I with metal debris particles in a 73-year-old male patient with an articulation of a ceramic head on the inner side of a cup with disturbed inlay. The cloud is at the left bottom close to the NOISE area and the distribution is “L”-shaped. The measured “cell count” was 6700 cells/µL.

**Figure 4 antibiotics-10-00346-f004:**
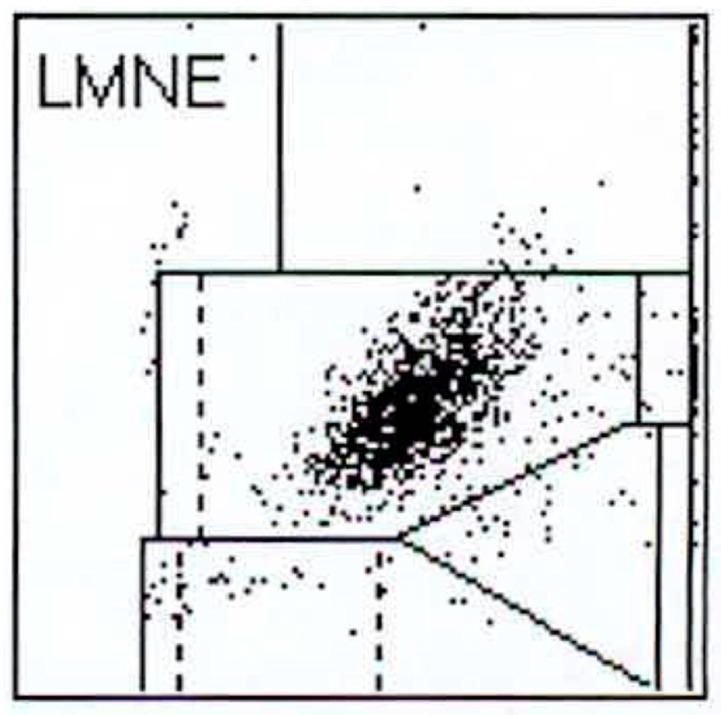
LMNE matrix of a type II (infection type) with a cloud in the area of the neutrophil leukocytes in a 75-year-old patient with a late periprosthetic joint infection of a total knee arthroplasty. The measured cell count was 1840 cells/µL.

**Figure 5 antibiotics-10-00346-f005:**
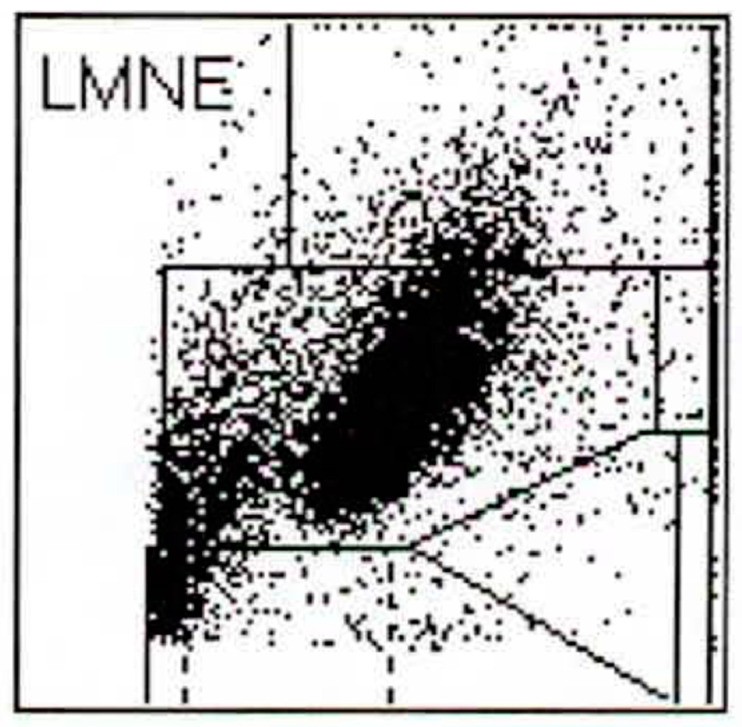
LMNE matrix of a type III (combined type) with one cloud in the area of the neutrophil leukocytes and a second cloud in the NOISE area in a 76-year-old male patient with a periprosthetic joint infection of a total knee arthroplasty. The measured cell count was 5840 cells/µL.

**Figure 6 antibiotics-10-00346-f006:**
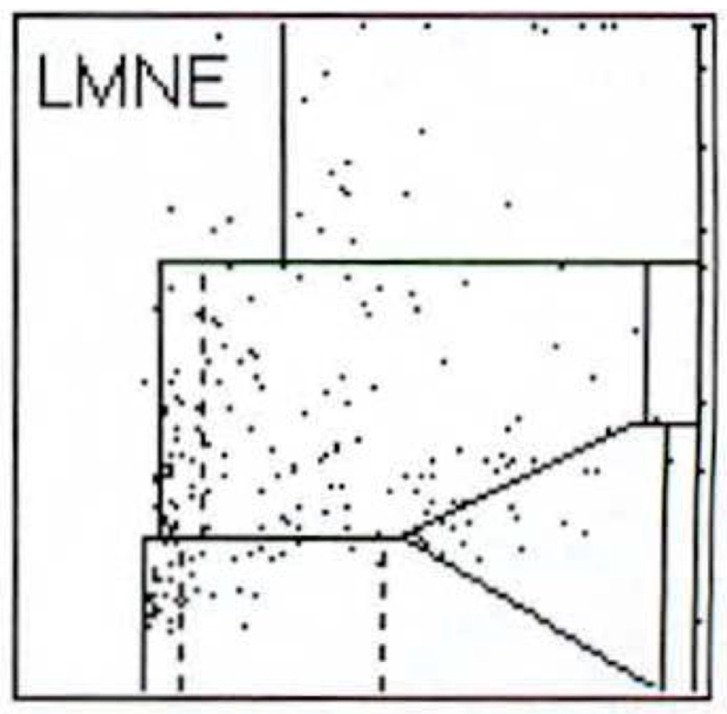
LMNE matrix of a type IV (indifference type) with no clear cloud or increase in cell types or particles in a 73-year-old patient. The measured cell count was 240 cells/µL.

**Figure 7 antibiotics-10-00346-f007:**
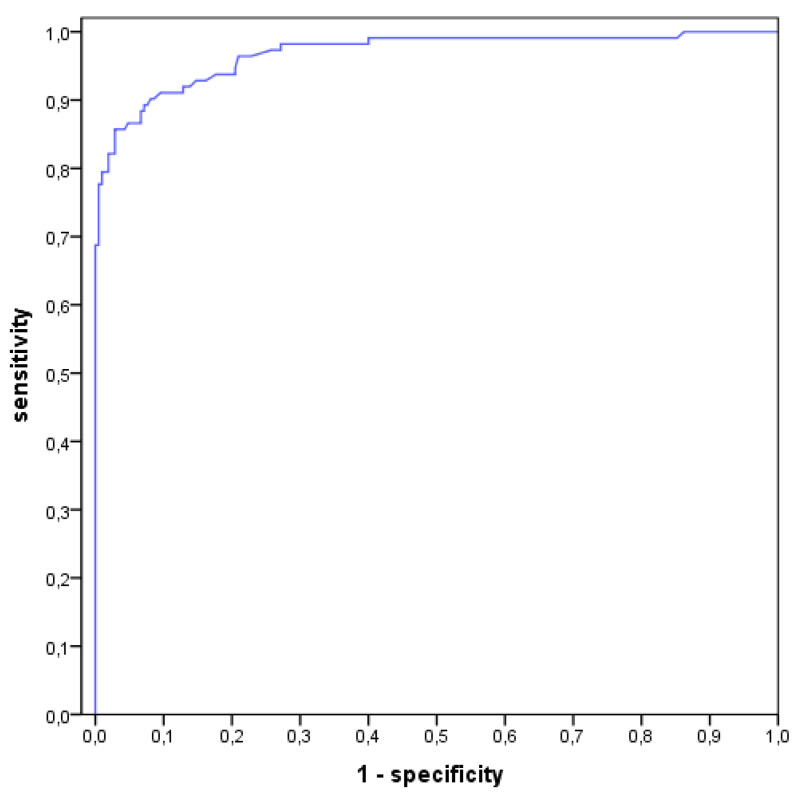
Receiver operating characteristics curve (ROC-curve) with the calculation of the threshold of cell count at a value of 1400 cells/µL with a sensitivity of 90.2% and specificity of 91.9%.

**Table 1 antibiotics-10-00346-t001:** Overview of the literature of cell count analysis in the aspirate for the diagnosis of periprosthetic joint infection. N = number of joints, H = hip arthroplasty, K = knee arthroplasty, 2 w = duration of symptoms of two weeks, PPV = Positive Predictive Value, NPV = Negative Predictive Value.

Autor	N	Cut-Off	Sensi-Tivity	Specifi-City	PPV	NPV	Accu-Racy
Balato 2018 [25]	167 K	>2800/µL >72% PMN	83.8% 84%	89.7% 91%			
Bergin 2010 [26]	64 K	>2500/µL >60% PMN	71%	98%	91%	93%	92%
Della Valle 2007 [7]	105 K	>3000/µL >65% PMN	100%	98.1%	97.6%	100%	98.9%
Ghanem 2008 [27]	429 K	>1100/µL >64% PMN	90.7% 95.0%	88.1% 94.7%	87.2% 91.6%	91.5% 96.9%	
Mason 2003 [28]	86 K	>2500/mL >60%PMN	98%	95%	91%	82%	
Parvizi 2006 [29]	145 K	>1760/µL >73%PMN					
Trampuz 2004 [8]	133 K	>1700/µL >65% PMN	94% 97%	88% 98%	73% 94%	98% 99%	
Zmistowski 2012 [30]	150 K	>3000/µL >75%PMN	93% 93%	94% 83%	93% 84%	94% 93%	93% 88%
Choi 2016 [12]	138 H	>5750/µL ≤ 2 w >1556/µL > 2 w	94% 91%	100% 94%	100% 87%	89% 97%	99% 95%
De Vecchi 2018 [31]	21 H + 45 K	>1600/µL >3000/µl	100% 93.7%	82.3% 91.2%	84.2% 90.9%	100% 93.9%	
Dinneen 2013 [32]	75 H	>1580/µL >80% PMN	89.5% 89.7%	91.3% 86.6%			
Higuera 2017 [33]	453 H	>3966/µL >80% PMN	89.5% 92.1%	91.2% 85.8%	76.4% 59.3%	97.5% 98.0%	93.0% 87.0%
Spangehl 1999 [34]	202 H	>5000/µL >80% PMN	89%	85%	52%	98%	
Schinsky 2008 [35]	201 H	>4200/µL >80% PMN	84% 84%	93% 82%	81% 65%	93% 93%	90% 83%

**Table 2 antibiotics-10-00346-t002:** Distribution of the patients according to the four different LMNE-matrices and the histological types described by Morawietz and Krenn [36,37,38].

LMNE-Type	Histological Classification				
	TYPE I	TYPE II	TYPE III	TYPE IV	TOTAL
LMNE-Type I	65	0	1	25	91
LMNE-Type II	5	68	5	2	80
LMNE-Type III	15	21	6	8	50
LMNE-Type IV	36	0	2	63	101
TOTAL	121	89	14	98	322

**Table 3 antibiotics-10-00346-t003:** Diagnostic value of the cell count at different thresholds (X) combined with the LMNE Type 2 or 3 (PJI); PPV = Positive Predictive Value, NPV = Negative Predictive Value., likelihood ratio green dark = superior diagnostic evidence, light green = high diagnostic evidence.

Threshold of Cell Count					Diagnostic	Value	Likelihood Ratio Positive	Likelihood Ratio Negative
		PJI			Accuracy	93.5%		
		yes	no		Sensitivity	98.2%	10.86	0.02
X = 500	pos.	110	19	129	Specificity	91.0%		
	neg.	2	191	193	PPV	85.3%		
		112	210	**322**	NPV			
		PJI			Accuracy	93.2%		
		yes	no		Sensitivity	93.8%	13.13	0.07
X = 1000	pos.	105	15	120	Specificity	92.9%		
	neg.	7	195	202	PPV	87.5%		
		112	210	**322**	NPV			
		PJI			Accuracy	93.8%		
		yes	no		Sensitivity	90.2%	21.04	0.10
X = 1500	pos.	101	9	110	Specificity	95.7%		
	neg.	11	201	212	PPV	91.8%		
		112	210	**322**	NPV			
		PJI			Accuracy	93.2%		
		yes	no		Sensitivity	86.6%	25.98	0.14
X = 2000	pos.	97	7	104	Specificity	96.7%		
	neg.	15	203	218	PPV	93.3%		
		112	210	**322**	NPV			
		PJI			Accuracy	93.8%		
		yes	no		Sensitivity	84.8%	59.38	0.15
X = 2500	pos.	95	3	98	Specificity	98.6%		
	neg.	17	207	224	PPV	96.9%		
		112	210	**322**	NPV			
		PJI			Accuracy	93.2%		
		yes	no		Sensitivity	82.1%	86.25	0.18
X = 3000	pos.	92	2	94	Specificity	99.0%		
	neg.	20	208	228	PPV	97.9%		
		112	210	**322**	NPV			

## Data Availability

The data presented in this study are available on request from the corresponding author. The data are not publicly available due to privacy.

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
