# Peer review of "The Graphical Representation of Cell Count Representation: A New Procedure for the Diagnosis of Periprosthetic Joint Infections"

_antibiotics, 2021, doi:10.3390/antibiotics10040346_

Round 1

Reviewer 1 Report

Far too many grammatical errors and run on sentences--this makes the intro very difficult to understand, please improve

Provide explanation of what graphical determination is, in a single line, within the introduction for some reference

The paper is a bit technically heavy, and the English is not the best--from what I can make out, the authors are using a diagnostic method by which they can separate various abrasion particles (which can artificially affect the leukocyte count number in the joint aspirate that is obtained) from true leukocytes and can represent this in a graphical fashion, allowing for a quick assessment of whether the joint WBC count obtained is truly suggestive of a PJI or of a noninfectious process.

Here is the problem with the premise of this paper however--joint fluid leukocyte count is only one of many criteria that can be used in helping to diagnosis PJI--the authors themselves confirm that joint leukocyte count is only a single minor criteria in the dx of PJI. Given this, I am having trouble understanding why anyone would go through all of this to help clarify the results of a joint leukocyte count--testing the fluid by culture and Gram stain, obtaining ESR/CRP, alpha defensin testing and assessing the joint clinically for the presence of sinus tract, etc. can all be instituted much more quickly in helping to establish a late PJI dx. In addition, even if this above technology was done and did not raise as much concern for a true joint leukocyte elevation, most surgeons would likely take the patient to the OR regardless to investigate the joint and more definitively rule out infection, if symptoms concerning for PJI were present. There is variability in the joint leukocyte counts because it is one of many tools that can be used to raise concerns for PJI--however it alone is not diagnostic for PJI. The authors need to explain exactly how this technology and graphic representation would be employed in the diagnosis of PJI, and if there is no practical application for it, then the utility of the testing is low and I don’t see a reason to perform it.

I hope I understood the premise of their paper well, it was unfortunately difficult to understand as the specifics of the graphical presentation and separation of abrasion particles vs leukocytes was not very clear to me. They would need to heavily revise lines 69-82 in particular as their explanation of the technology and process is not clear (differences between the infection types in the clinical setting, what is LMNE, what is NOISE, etc).

The abstract also has many run on sentences and spelling errors that need to be fixed. Lastly, the order of the paper (results, methods, graphs, tables, discussion, etc) seem to be out of order--it should be abstract, then intro, then methods, then results, then conclusions, then discussion, then tables/charts. Perhaps this is due to formatting, but I am not sure.

The authors should explain their technology and the precise utility of this in improving the diagnosis of PJI

Author Response

Thank you very much for reviewing our paper. All comments are adressed in new version of the manuscript and the changes are made in red.

  • The English language is improved by a native speaker and scientist.
  • In the introduction section is clarified why this new technique is helpful for the differentiation between abrasion particles and real leukocytes with periprosthetic joint infection even WBC counting is only one minor criteria. Please also read the comments of reviewer 2.
  • The technique is explained better. Moreover, we changed the numbering of the figures. Figure 1 show the LMNE-matrix which also describe the NOISE-area. This was numbered as Figure 7 in the old manuscript, which made it more difficult to understand.
  • The order of the manuscript is according to the guidelines

Reviewer 2 Report

The manuscript of Bernd Fink et al. deals with the study of graphic representation of the cell count analysis of synovial aspirates of joints with endoprostheses. Manuscript describes a new and helpful method for determining the cell count of the synovial aspirate between real periprosthetic infections with increased leukocyte count and incorrectly increased numbers due to particle abrasion. All the analysis of synovia was estimated using laboratory diagnostic device ABX Pentra XL 80. Those results increase the diagnostic value of the cell count analysis. Moreover, charts and tables could be proven useful for researchers in the field (especially table 2-3). Therefore, I would recommend these articles for publication after clarifications of following questions. The publication is well written but to me there are lacks that should be solved/explained:

Could authors provide scale (cell size) for figures 1-5? The quality of these figures should be improved.

The only method on which authors elaborate is cell types graphically mapped (LMNE matrix) depending on their cell volume (x-axis) and their light scattering or refraction and absorption (y-axis). Authors should describe also other methods to differentiate between joint infection and metallic abrasion particles. Did the authors measure the presence of metallic particles in the samples by other methods?

All considered, I would strongly recommend this article for publication in the journal "Antibiotics".

Author Response

Thank you very much for reviewing our paper. All comments are adressed in new version of the manuscript and the changes are made in red.

  • The figures printed out by the ABX Pentra XL 80 do not have scales. The have fields which are explained in figure 1 (named figure 7 in the old version of the manuscript). Matrices with scales are not available.
  • Unfortunally, the quality of the figures could not be improved because the matrices are printed out on a paper by the ABX Pentra XL 80 and thereafter were scanned. Because the matrices are quite small the magnification of these decrease the quality.
  • In the discussion section we describe other methods to differentiate between joint infection and metallic abrasion particles. In general we perform microscopic examination of the native aspirate. We did not add that in this study to prevent a bias in the interpretation of the LMNE-matrix-types. We can add that if the reviewer wish.

Round 2

Reviewer 1 Report

Much better explanation of significance of their work, English is much improved, although some of the

figures have minor spelling errors.